# Faster Gastric Emptying Is Unrelated to Feeding Success in Preterm Infants: Randomized Controlled Trial

**DOI:** 10.3390/nu11071670

**Published:** 2019-07-21

**Authors:** Maria Elisabetta Baldassarre, Antonio Di Mauro, Osvaldo Montagna, Margherita Fanelli, Manuela Capozza, Jennifer L. Wampler, Timothy Cooper, Nicola Laforgia

**Affiliations:** 1Neonatology and Neonatal Intensive Care Unit, Department of Biomedical Science and Human Oncology, University of Bari “Aldo Moro”, 70124 Bari, Italy; 2Medical Statistics, Department of Interdisciplinary Medicine, University of Bari “Aldo Moro”, 70124 Bari, Italy; 3Department of Medical Affairs, Mead Johnson Nutrition, Evansville, IN 47721, USA

**Keywords:** infant, premature, gastric emptying, infant formula, ultrasonography, feeding tolerance

## Abstract

Objectives: To evaluate the relationship between gastric emptying (GE) time and days to achievement of full enteral feeding (≥140 mL/kg/day) in preterm infants randomly assigned to receive one of two marketed study formulas for the first 14 feeding days: intact protein premature formula (IPF) or extensively hydrolyzed protein (EHF) formula. Methods: In this triple-blind, controlled, prospective, clinical trial, we report GE time (time to half-emptying, t_1/2_) by real-time ultrasonography on Study Day 14, in preterm infants receiving IPF or EHF formula. The association between GE time and achievement of full enteral feeding was evaluated by Pearson correlation. Per-protocol populations for analysis included participants who (1) completed the study (overall) and (2) who received ≥ 75% study formula intake (mL/kg/day). Results: Median GE time at Day 14 was significantly faster for the EHF vs. IPF group overall and in participants who received ≥ 75% study formula intake (*p* ≤ 0.018). However, we demonstrated GE time had no correlation with the achievement of full enteral feeding (*r* = 0.08; *p* = 0.547). Conclusion: Feeding IP premature formula vs. EH formula was associated with shorter time to full enteral feeding. However, faster GE time did not predict feeding success and may not be a clinically relevant surrogate for assessing feeding tolerance.

## 1. Introduction

When a mother’s own breast milk or donor breast milk is not available, there is a desire to select infant formulas that would minimize both fasting and the time to achieving full enteral nutrition while considering the potential for necrotizing enterocolitis (NEC) and related morbidities often associated with feeding difficulties. Many preterm infants may experience feeding intolerance, as evidenced by vomiting, gastric residuals (probably reflecting impaired gastric emptying (GE)), and abdominal distention or perceived constipation which are all likely related to immature gastrointestinal motility [1]. Recognizing the disadvantages associated with periods of fasting and inadequate enteral nutrition must be balanced with interpreting signs of feeding intolerance. The impact on GE is often considered an important factor in the choice of infant formulas in preterm infants. Infant formulas with demonstrated faster GE are often selected [2,3,4,5] in clinical practice despite a lack of clear association with improved clinical outcomes.

Because of the perception of improved tolerance outcomes, the use of hydrolyzed protein formulas has increased in popularity over the past two decades. Time to GE is also influenced by other factors including the type of protein, amount and type of fatty acids, energy density, fiber content, probiotic, and osmolality [6,7,8,9,10,11]. Implicit in interpreting results of GE studies has been the assumption that faster GE will result in earlier achievement of full enteral feeding in premature infants. However, previously published reports regarding this relationship are mixed, with earlier achievement of full enteral feedings reported as either associated [12] or not associated [10,13] with feeding extensively hydrolyzed formulas.

In the current pilot study, we evaluate the correlation between GE time and the achievement of full enteral feeding (defined as a daily intake of ≥ 140 mL/kg/day) in preterm infants randomly assigned to receive one of two cow’s milk-based study formulas over the first 14 days of feeding: intact protein premature infant formula (marketed Enfamil^®^ Premature, Mead Johnson Nutrition (MJN), Evansville, IN, USA) or extensively hydrolyzed protein infant formula (marketed Pregestimil^®^, MJN, Evansville, IN, USA).

## 2. Methods

### 2.1. Study Design and Participants 

Participants were enrolled in a triple-blind, randomized, controlled prospective pilot study to evaluate the achievement of full enteral feeding (daily intake of ≥ 140 mL/kg/day) in premature infants randomized to receive one of two study formulas over the first 14 days of feeding: intact protein premature infant formula (IPF: marketed Enfamil^®^ Premature, Mead Johnson Nutrition (MJN), Evansville, IN, USA) or extensively hydrolyzed protein infant formula (EHF: marketed Pregestimil^®^, MJN, Evansville, IN, USA). Human milk feeding was encouraged; assigned study formulas were used when mother’s own milk was not available.

Eligible infants were enrolled between February 2014 and February 2016 in the Neonatal Intensive Care Unit (NICU) of University of Bari “Aldo Moro”, Bari, Italy. Trial registration: ClinicalTrials.gov: NCT01987154

The research protocol and informed consent forms observing the Declaration of Helsinki (including October 1996 amendment) were approved by the “Azienda Ospedaliero-Universitaria Consorziale Policlinico” Independent Ethics Committee, Bari, Italy. The study complied with good clinical practices. Parents or a legally authorized representative provided written informed consent prior to enrollment.

Inclusion criteria were: (1) 28–33 completed weeks’ gestational age at enrollment, (2) birth weight of ≥700 to 1750g (3) appropriate weight for GA (AGA, defined as birth weight between and inclusive of the 10th and 90th percentiles using the Italian Neonatal Study growth chart [14]). Furthermore, infants were enrolled if (4) they never received enteral feedings prior to randomization or (5) if they had enteral intake <30 mL/kg/day, by or before first 24 h from first enteral feeding.

Exclusion criteria were: (1) history of underlying metabolic or chronic disease, (2) gastrointestinal diseases or malformations, (3) cystic fibrosis and other genetic diseases, (4) major surgery, (5) unstable blood pressure, (6) Grade III or IV intraventricular hemorrhage (IVH), (7) ventilator dependency or requiring > 40% Fi02, and (8) 5 min Apgar score <4. 

Daily enteral intake (mL; mean ±SD) was calculated for each participant.

Both IPF and EHF study formulas were provided at the same energy content (24 kcal/fl oz) and had similar osmolality (IPF: 310 vs. EHF: 340 mOsm/kg H_2_O), but differed in whey:casein content (IPF: intact 60:40 ratio of whey to casein; EHF: casein hydrolysate only) and percentage (38% vs. 53%) of total fat as medium chain triglyceride (MCT) oil. 

### 2.2. Assessment of Gastric Emptying by Real-Time Ultrasonography

Gastric emptying rate was assessed by real-time ultrasonography method as previously described [15]. Briefly, the evaluation was performed in the same environment at the same time in the morning. Participants were examined in a supine position with an abdominal transducer (DC-8 Diagnostic Ultrasound System, Mindray, North America). Between examinations, the subjects remained supine in a neonatal incubator. The same operator (Dr. Osvaldo Montagna) assessed all enrolled infants and was masked regarding participant clinical data. Gastric emptying was monitored indirectly by determining the longitudinal (*D*_1_) and anteroposterior (*D*_2_) cross-sectional diameters of a single section of the gastric antrum, which was measured at the longitudinal scan at the level of the abdominal aorta and the left lobe of the liver. At each observation, three measurements were taken using the mean values of the longitudinal (*D*_1mean_) and anteroposterior (*D*_2mean_) diameters to calculate the gastric antral area. At this level, the scan showed the stomach shaped as either a circle or an ellipse; therefore, the gastric antral cross-sectional area (*A*) was calculated in all subjects using the formula:A = π × longitudinal radius × anteroposterior radius → A = π × (D1mean/2) × (D2mean/2) → A = π × (D1mean × D2mean/4)(1)

The intragastric volume was assumed to be directly proportional to the cross-sectional area of the gastric antrum. Measurements were taken before the milk bolus (0 min) and 15, 30, 45, 60, 75, and 90 min after the bolus. The test bolus consisted of: a single feeding (either IPF or EHF study formula) providing the same quantity received in 3 hours if continuous feeding or the same quantity received in a single bottle if bolus feeding, during Study Day 13. 

The maximum increase in gastric antral cross-sectional area from fasting to the end of the meal (AA_max_) was calculated as follows:AA_max_= *A*_max_ − *A*_f_(2)
where *A*_max_ is the antral area computed at the end of the meal (15 min), and *A*_f_ is the antral area computed at fasting (0 min). 

The difference between the value observed at each observation (*A*_n_) and the one computed during fasting (*A*_f_) was calculated and termed:AA_n_ (AA*n* = *A*_n_ − *A*_f_)(3)
The Gastric Emptying Rate was estimated assuming AA_max_ was 100% (AA_max_/AA_max_ × 100); all the values calculated from each observation were then transformed as a percentage decrease in AA_max_ as follows: AA_n_/AA_max_ × 100(4)

The GE time (time to half-emptying, *t*_1/2_) was defined as the time when the Gastric Emptying Rate had reached 50%, and computed by a linear interpolation [16].

### 2.3. Statistical Analysis

GE time (time to half-emptying, t_1/2_) curves using Kaplan-Meier estimates were compared between groups by Log Rank Test. Pearson correlation was used to evaluate the relationship between GE time and days to full enteral feeding. Populations for the evaluation established per protocol included (1) primary: participants who met study entrance criteria and completed study feeding, and (2) subset analysis: of those who completed study feeding, those participants who received ≥75% enteral intake from the study formula (average over total study days; mL/kg/day). All observations of participants who had not reached full enteral feeding by Day 14 or prior to experiencing 4 consecutive days with no enteral intake were treated as a censored observation on the Study Day prior to this 4-day period. All testing was conducted at α = 0.05. All analyses were conducted using IBM^®^ SPSS^®^ Statistics 23.

## 3. Results

### 3.1. Study Population

Between February 2014 and May 2016, 65 premature infants in the Neonatal Intensive Care Unit (NICU) of University of Bari “Aldo Moro”, Bari, Italy were enrolled and randomized; 60 completed the study per protocol (IPF: 30; EHF: 30) and of these, 23 received ≥75% study formula intake (IPF: *n* = 11; EHF: *n* = 12; *p* = 0.79) (Figure 1).

Infant characteristics including birth anthropometric measures at baseline as well as gender, birth type, Apgar score, and gestational age were similar between groups (Table 1).

### 3.2. Gastric Emptying (GE) Time

GE time at Day 14 was significantly faster in EHF vs. IPF by primary analysis (median t_1/2_: 54 min vs. 59 min; *p* = 0.031) (Figure 2A), as well as subset analysis for participants receiving ≥75% enteral intake from study formula (median t_1/2_: 53 min vs. 62 min; *p* = 0.018) (Figure 2B). 

### 3.3. Effect of GE Time on Achievement of Full Enteral Feeding

There was no correlation of the effect of GE time on achievement of full enteral feeding for participants by primary analysis (*r*^2^ = 0.08; *p* = 0.547) (Figure 3) or by subset analysis for participants receiving ≥75% enteral intake from study formula (*r*^2^ = 0.05; *p* = 0.825).

## 4. Discussion

In the current pilot study, we aimed to determine the relationship between time to gastric emptying and achievement of full enteral feeding in preterm infants, a nutritional milestone for preterm infants. We report here that faster gastric emptying did not predict feeding success in preterm infants. We demonstrated that GE time had no correlation with the achievement of full enteral feeding in all participants who completed the study and in preterm infants receiving ≥75% of enteral intake from study formula. Consequently, results in the current study prompt a re-examination of informal guidelines for EH formula use in preterm infant feeding as well as the clinical relevance of using gastric emptying as a surrogate marker of feeding tolerance.

In the current study, real-time ultrasonography demonstrated faster time to gastric emptying in preterm infants receiving an EH infant formula vs. an IP premature formula. Previous studies have also demonstrated reduced time to gastric emptying in term and preterm infants receiving infant formula with EH vs. IP protein [17,18,19]. Real-time ultrasonography, a safe and non-invasive technique, provides repeatable results that correlate well with scintigraphy, which is considered the gold standard to assess gastric transit [20]. Gastric emptying may also be affected by other infant formula constituents and properties, including energy content [8,21], osmolality [6,8,22,23], and different whey:casein content [11,17,24,25]. As noted, the current study formulas had the same energy content and similar osmolality, but differed in whey:casein content and percentage of medium chain triglyceride (MCT) oil. No consistent effect of differences in MCT content on gastrointestinal transit has been demonstrated [26]. More specifically, no significant group differences were reported in two small studies of premature infants [27,28], whereas a much older study demonstrated greater inhibition of transit associated with longer chain fatty acids compared to MCT [29]. Although the study formulas in the current trial included some differences in addition to the protein source and MCT, both marketed formulas reflected real-world choices available to neonatal units. 

Most strikingly, no correlation between gastric emptying time and achievement of full enteral feeding was demonstrated for participants receiving an IP or EH study formula. Previous studies have also demonstrated no relationship between feeding in IP vs. EH formulas and other outcomes considered measures of tolerance: the number of episodes of vomiting; frequent, elevated gastric residual volumes; or number episodes of fasting [13,30,31]. Apart from a single 2011 study (50 preterm infants 25–32 weeks’ GA at birth) [32], we are aware of no published evaluations of the relationship between gastric emptying and full enteral feeding. In the 2011 report, no clinical measures (including GE time, gastric residual volumes, lactase activity, gastrointestinal permeability, and fecal calprotectin concentration) reliably predicted time to attainment of full gavage feeding. In NICU practice, large gastric residuals or slow emptying often result in either holding of feeding or a reluctance to advance volume, thereby secondarily prolonging time to achieve full enteral feedings. Increasingly, neonatologists have abandoned the evaluation of gastric residual volumes as a reliable prediction of pathology due to little data demonstrating increased success in feeding advancement [33,34]. Results of the current study suggest using GE time to predict achievement of full enteral feeding is also unreliable. Contrary to expectations, the current study suggests that GE time bears no relationship to success at achieving full enteral feedings.

## 5. Conclusions

The results of this study demonstrated that GE time had no correlation with achievement of full enteral feeding in preterm infants receiving an EH infant formula compared to an IP premature formula during the first 14 days of feeding either overall or in preterm infants receiving ≥75% study formula. As part of this study, we reported that the achievement of full enteral feeding, a nutritional milestone, was similar between groups overall and significantly shorter for participants receiving ≥75% of enteral intake from an IP premature infant formula. Current interpretation regarding the role for EH formulas in preterm infants in achieving full enteral nutrition, reducing the time of parenteral nutrition, and using GE time as a surrogate measure of feeding tolerance, bears re-examination. 

## Figures and Tables

**Figure 1 nutrients-11-01670-f001:**
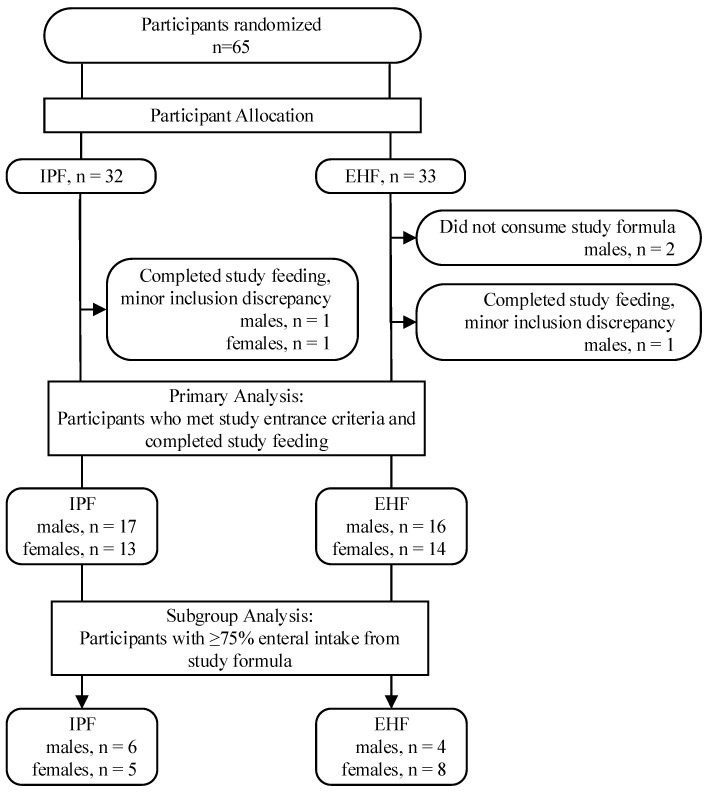
Study allocation.

**Figure 2 nutrients-11-01670-f002:**
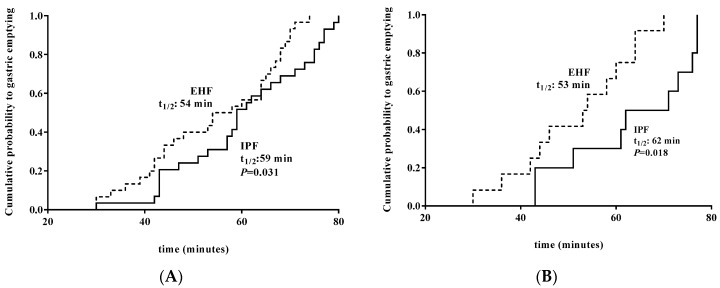
(**A**) Significantly faster GE time at Day 14 (minutes, median cumulative probability) in EHF vs. IPF in all participants who completed the study, t_1/2_: 54 min vs. 59 min; *P* = 0.031). (**B**) Significantly faster GE time at Day 14 (minutes, median cumulative probability) in EHF vs. IPF in participants receiving ≥75% of enteral intake from study formula, 53 min vs. 62 min; *P* = 0.018) (subset analysis). **EHF, dotted line; IPF, solid line**.

**Figure 3 nutrients-11-01670-f003:**
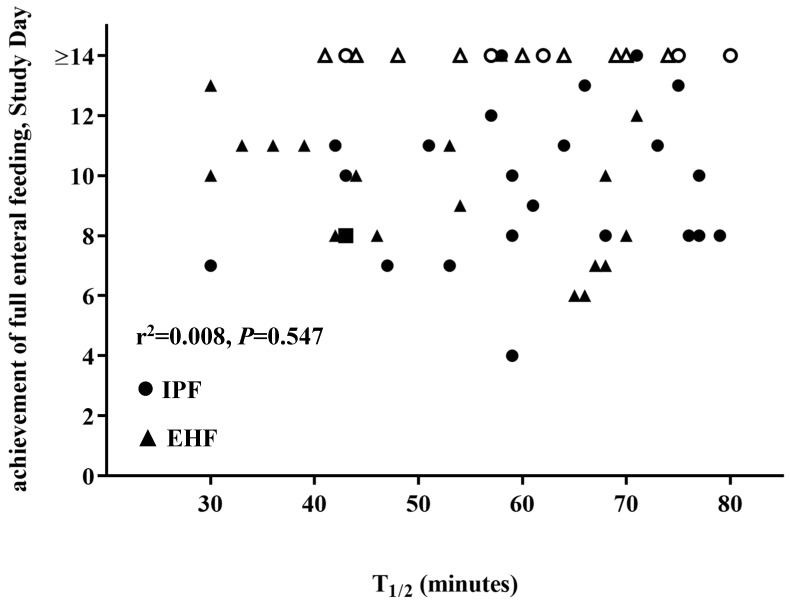
No correlation of the effect of GE time (T _1/2_, min) and achievement of full enteral feeding. Study Day for participants overall (*r*^2^ = 0.08; *p* = 0.547). IPF, ●; EHF, ▲ (unshaded symbols represent censored participants, i.e., did not reach full enteral feeding by Study Day 14). > 14 days data have been censored and excluded from correlation analysis.

**Table 1 nutrients-11-01670-t001:** Infant characteristics at study entry.

	Study Group		*p*
	IPF	EHF	
gender, *n* (%)			
male	17 (56.7)	16 (53.3)	0.795
female	13 (43.3)	14 (46.7)	
birth type, *n* (%)			
singleton	15 (50)	17 (56.7)	0.605
twin	15 (50)	13 (43.3)	
Cesarean Section, *n* (%)	26 (86.7)	26 (86.7)	1.000
Apgar score, *n* (%)			
6–7	3 (10.0)	6 (20)	0.760
8	6 (20.0)	6 (20)	
9	16 (53.3)	12 (40)	
10	5 (16.7)	6 (20)	
gestational age (days) *	30.1 (1.6)	30.9 (1.9)	0.803
birth anthropometrics *			
weight (g)	1278.7 (259.7)	1301 (293.2)	0.756
length (cm)	38.1 (2.8)	38.1 (3.3)	0.973
head circumference (cm)	27.3 (1.7)	27.7 (2.4)	0.481

* Mean ± standard deviation (SD).

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
