# Peer review of "Faster Gastric Emptying Is Unrelated to Feeding Success in Preterm Infants: Randomized Controlled Trial"

_nutrients, 2019, doi:10.3390/nu11071670_

Reviewer 1 Report

It would be useful to the readers the inclusion of reason(s) why preterm infants did not receive human milk. Is there any correlation between GE and protein content of the formula? I wonder whether data should be normalized to protein content or not.

Author Response

It would be useful to the readers the inclusion of reason(s) why preterm infants did not receive human milk. 

Human milk feeding was encouraged; assigned study formulas were used when mother’s own milk was not available. We better explain this in methods section (line 75-76) and add a study flow chart in results (figure 1).

Is there any correlation between GE and protein content of the formula? I wonder whether data should be normalized to protein content or not.

We did not normalized the data to protein content. However the difference between the two marketed study formulas was of 0,2 g. In detail:  3,0 g , 12% calories in IPF study formula and 2,8 g/11% calories in EHF study formula.

Reviewer 2 Report

Thanks for the opportunity to review this manuscript.

In this paper, Baldassare et al report finding of a study that was “piggy-backed” on a prospective clinical trial they performed to study achieved feeding volume and time to achieve full feeds in preterm infants receiving intact protein preterm formula or extensively hydrolyzed protein formula. In the original study (that has not been published yet and I believe it has been submitted to this journal at the same time) apparently showed shorter time to full
enteral feeding and higher achieved feeding volume in babies receiving intact protein formula. In the current paper they report that the gastric emptying (GE) time was shorter in babies receiving the extensively hydrolyzed and accordingly showed no correlation with the achievement of full enteral feeding.

The paper is interesting. The question is highly relevant for the practicing neonatologist; the main result (the shorter GE time in case of hydrolyzed formula does not mean shorter time to full feeds) is easy to interpret.

The main problem with the manuscript is that its methodology and findings are closely linked with those of the main paper (presumably also submitted to this journal). That lead to two problems:

- This reviewer had no access to the other manuscript and without that the evidence presented this paper was hard to evaluate.

- The authors discuss some of the findings of the other paper as “results” in this paper as well. However, without all the details and evidence presented, those results “hang in the air”. I would suggest that for the sake of clarity, the authors separate the findings of two manuscripts from each other even if they appear in the same edition of the same journal as these papers will be available as separate references and pdfs.

Specific comments:

Abstract:

Line 16: “we previously reported” : the other paper has not been published yet, it is being peer reviewed.

Introduction:

Well written and clear

Methods:

Please give more descriptive statistics (e.g. mean, SD), about the subjects gestational age, postnatal age, birth weight, actual age, weight percentile. Also provide data about gender, Apgar score etc. and show that the groups were not significantly different in these charateristics. I think this would be best done in a table.

The authors list the exclusion criteria but the sentence is long and difficult to understand (e.g. had they grade 3 -4 IVH or not. Please list inclusion and exclusion criteria in a more structured way.

Line 93, reference 15: is this the correct reference? This refers to a study of healthy adult volunteers taking bread.

Line 105: I suspect “p” refers to “pi” (3.14…), please make it clear.

Line 125: Please provide evidence that gastric emptying is a linear process and therefore it is justified to use linear interpolation.

Line 128: are the assumptions of Pearson’s correlation (absence of outliers, normality of variables, linearity, and homoscedasticity) fulfilled. If not, consider other tests, eg. Spearman correlation based on ranking.

Results:

Line 139: “As previously reported” – refers to the other paper which is under peer review

Line 143-144 – provide more data about these characteristics as suggested in an earlier comment.

Line 146-151: These results belong to the other paper and should not be discussed here.

Discussion:

Overall balanced and well written

Conclusion

Line 204-205: the authors do not provide evidence for this statement in the paper.

Figures are generally clear, please mention in Figure 2 legend that the >14 day data have been censored and excluded from correlation analysis

Author Response

In this paper, Baldassare et al report finding of a study that was “piggy-backed” on a prospective clinical trial they performed to study achieved feeding volume and time to achieve full feeds in preterm infants receiving intact protein preterm formula or extensively hydrolyzed protein formula. In the original study (that has not been published yet and I believe it has been submitted to this journal at the same time) apparently showed shorter time to full
enteral feeding and higher achieved feeding volume in babies receiving intact protein formula.

In the current paper they report that the gastric emptying (GE) time was shorter in babies receiving the extensively hydrolyzed and accordingly showed no correlation with the achievement of full enteral feeding.

The paper is interesting. The question is highly relevant for the practicing neonatologist; the main result (the shorter GE time in case of hydrolyzed formula does not mean shorter time to full feeds) is easy to interpret.

Thank you for your interest and for your constructive comments.

The main problem with the manuscript is that its methodology and findings are closely linked with those of the main paper (presumably also submitted to this journal). 

That lead to two problems:

- This reviewer had no access to the other manuscript and without that the evidence presented this paper was hard to evaluate.

- The authors discuss some of the findings of the other paper as “results” in this paper as well. However, without all the details and evidence presented, those results “hang in the air”. I would suggest that for the sake of clarity, the authors separate the findings of two manuscripts from each other even if they appear in the same edition of the same journal as these papers will be available as separate references and pdfs.

We agree with this comment. 

We previously asked to Editorial Office to assign both manuscript to the same Reviewers, but they didn’t. 

However, we accordingly separate the findings of the two manuscript from each other in this version.

Specific comments:

Abstract:

Line 16: “we previously reported” : the other paper has not been published yet, it is being peer reviewed.

We accordingly modify the paragraph

Introduction:

Well written and clear

Thank you

Methods:

Please give more descriptive statistics (e.g. mean, SD), about the subjects gestational age, postnatal age, birth weight, actual age, weight percentile. Also provide data about gender, Apgar score etc. and show that the groups were not significantly different in these charateristics. I think this would be best done in a table.

We accordingly add a table with demographic informations.

The authors list the exclusion criteria but the sentence is long and difficult to understand (e.g. had they grade 3 -4 IVH or not. Please list inclusion and exclusion criteria in a more structured way.

We accordingly list inclusion and exclusion criteria in a structured way: line 86-94

Line 93, reference 15: is this the correct reference? This refers to a study of healthy adult volunteers taking bread.

The ultrasound tecnique to assess the gastric emptying (GE) time is the same for newborn and adults. The reference is correct.

Line 105: I suspect “p” refers to “pi” (3.14…), please make it clear.

It refers to “p”. We accordingly correct

Line 125: Please provide evidence that gastric emptying is a linear process and therefore it is justified to use linear interpolation.

We add the reference.

Line 128: are the assumptions of Pearson’s correlation (absence of outliers, normality of variables, linearity, and homoscedasticity) fulfilled. If not, consider other tests, eg. Spearman correlation based on ranking.

Yes, they are fulfilled.

Results:

Line 139: “As previously reported” – refers to the other paper which is under peer review

We accordingly separate the findings of the two manuscript from each other in this version.

Line 143-144 – provide more data about these characteristics as suggested in an earlier comment.

We accordingly added table 1.

Line 146-151: These results belong to the other paper and should not be discussed here.

We accordingly separate the findings of the two manuscript from each other in this version.

Discussion:

Overall balanced and well written

Conclusion

Line 204-205: the authors do not provide evidence for this statement in the paper.

We accordingly delete this statement.

Figures are generally clear, please mention in Figure 2 legend that the >14 day data have been censored and excluded from correlation analysis

We corrected figure legend.